# Electric Vehicle Penetration in Distribution Network: A Swedish Case Study

**Henrik Maninnerby [1], Sune Bergerland [2], Stavros Lazarou [3] and Andreas Theocharis [4,\*]**

[1]  Studentwork AB, 112 18 Stockholm, Sweden
[2]  Karlstads El- och Stadsnät AB, 651 84 Stockholm, Sweden
[3]  Department of Electrical and Electronic Engineering Educators, School of Pedagogical & Technological Education (ASPETE), 141 21 Heraklion, Greece
[4]  Department of Electrical Engineering, Karlstad University, 651 88 Karlstad, Sweden
[\*]  Correspondence: andreas.theocharis@kau.se

**Abstract:** This study aims to simulate the use of renewable energy in the form of different energy sources, such as solar cells, district heating, and in the presence of battery storage and for high penetration of electric vehicles in a typical Swedish power grid. The EnergyPLAN software is used. The purpose is to examine the demands in order to cope with the needs that may arise and to create a better understanding of how renewable energy affects the power balance and future investments in the case of a typical Swedish distribution system. The importance of this research is mainly based on the fact that it represents a real network, as it operates today, which is analyzed using the expected electric vehicle penetration. The aim is to investigate the expansion needs for maintaining the current quality for service despite the addition of new loads. In addition, the regional and national special regulatory and operational requirements are taken into account and described in this work.

**Keywords:** renewable energy sources; electric vehicles; EnergyPLAN; distribution grid; distributed generation

## 1. Introduction

As a consequence of the United Nations' (UN) latest agreement in Paris in December 2015 on climate changes [1,2] as well as due to the rapidly public increasing interest on Electric Vehicles (EVs), a number of European countries, including Sweden, and some states in the USA agreed to expand the market for EVs as an effort to reduce the amount of greenhouse gases [3]. As analyzed by Morais et al. [4], if the number of EVs increases dramatically in Sweden, the electricity system could be possibly burdened. The problem might be even more intensive in a particular case where the owners of EVs would like to charge with higher currents than 10 A single-phase charging, which is typical today for a Swedish large customer. These barriers are better explained by Knezovic et al. [5]. An extensive assessment of urban networks is provided by Fichera et al. [6]. However, EVs could provide ancillary power system services alleviating constraints [7] and congestions [8].

In Sweden, electricity consumption is mostly affected by cold winters, especially in areas where space heating is based on electrical units. In particular, in Karlstad, where the presented case study takes place, only about 5% of the households are not heated by electrical units. Sweden's power companies are regulated by the Energy Market Inspectorate, "*Energimarknadsinspektionen*", to ensure that fees paid by the power companies' customers are maintained at a reasonable level [9].

According to Statistics in Sweden [10], the number of cars, buses, and trucks fully powered by electricity was around 290 at the end of 2008. If electric hybrid (EHs) vehicles were included, the corresponding number increases to approximately 13,800 vehicles, which could be connected to

the grid following, according to a study from de Durana et al. [11]. According to the same statistics, the amount of pure EVs was slightly more than 11,100 at the end of 2017 and, including EHs, the amount increases to more than 115,000. In 2017, almost 39,000 fully or partially EVs were registered in Sweden, which results in almost 34% of the total new registered vehicles. This indicates that the number of EVs and EHs will only increase over the next few years. Moreover, EHs contribute to the largest increment with almost 50% of the total over the last five years. Except from EVs, Plug-in EHs could also contribute to power system balancing [12].

It is of paramount importance for electrical grid owners, electrical grid operators, and all involved stakeholders to have an integrated knowledge with regard to future electrical power needs in order to establish a safe strategy for grid investments and new business models. As such, several future scenarios are explored in order to define suitable strategies toward a feasible solution with high penetration of renewable energy systems (RES), EVs, and energy storage devices (ESD) [13,14]. In Reference [13], the prospects for realization of the 100% renewable energy system in North Macedonia by making use of the EnergyPLAN model is investigated. In the first scenario, a 50% RES has been created for the year 2030 as a first step toward the 100% future RES. In the second scenario, a 100% RES in the year 2050 has been designed. In Reference [14], a study is presented toward a 100% RES for Ireland based on the EnergyPLAN, since it accounts for all sectors of the energy that need to be taken into account for the integration of renewable sources such as the electricity, the heat, and the transport sectors. The investigation has been conducted by developing three different 100% RES with each focusing on a different resource. Specifically, biomass, hydrogen, and electricity resources have been used and compared with reference to the existing Irish energy system. There are several similar investigations based on future scenarios toward the development of a 100% RES. Each particular case study is unique because it includes its own special characteristics and requirements.

The aim of this paper is to investigate several scenarios to achieve an electrical power balance under high penetration of EVs in the Swedish distribution system. Specifically, this study is based on a real system and its findings were applied to this network, instead of being applied to a generalized approach [15]. The investigated area is Henstad/Hultsberg in the northwest of Karlstad, Sweden, in the Karlstads El- och Stadsnät's power network. Several scenarios have been developed and simulated. The parameters that configure the scenarios are (i) the number of grids connected EVs, (ii) the level of charging rate, (iii) the charging profile, (iv) the flexible demand, and (v) the district heating versus heating using electrical units. Moreover, the case of residential PV systems is included when they are used for household demands. The purpose of the paper is to simulate the development of RES in the presence of different sources, such as PVs, district heating, and ESD. The presented analysis has been conducted by using the EnergyPLAN software in References [16] and [17], which offer a proven research tool. This particular city area has been selected for the study in order to investigate the vulnerability for electricity shortages in case of high penetration of EVs and distributed generation [18]. The purpose of the paper is to simulate the development of RES in the presence of different sources, such as PVs, district heating, and ESD. As such, the impact of high EVs penetration to the power network is elaborated and a better understanding of potential required measures and grid investments for the feasible operation of the RES are demonstrated.

## 2. Materials and Methods

### 2.1. The EnergyPLAN Software

EnergyPLAN simulates the operation of large-scale energy systems on an hourly basis, including the electricity, heating, cooling, industry, and transport sectors. The EnergyPLAN [16] is a simulation software for renewable energy integration studies, initially developed by Ph.D. Henrik Lund at Aalborg University in 1999. Since then, the program has been improved, updated, and, thus, expanded with many new features to its current version, 132, as several companies and other universities participate

in the project. Among other things, the main capabilities of the program are to assist in the design of new power networks as well as the analysis of the energy impact of different energy strategies.

The results of the simulations can be obtained in a few different ways. Through Run (Clipboard), the data is copied, which can be pasted into Microsoft Excel for further processing. The Run (Display) shows the same data in the form of a box that appears on the screen. Run (Print) prints the data on a specified printer and Run (Serial) allows running up to eleven values of RES to review the changes. For example, it is possible to see how different amounts of power from solar panels affect the total critical overproduction of electricity, fuel, exports and imports of electricity, $CO_2$, costs, and more. However, this function only shows the corresponding value without considering fluctuations during the year.

### 2.2. The Primary Data of the Simulation

This study has been conducted for Karlstads El- och Stadsnät, Henstad/Hultsberg area in the Karlstad power network. Initially, it was necessary to find out what transformers are in the area. Based on the data available from Karlstads El- och Stadsnät and searching streets using local maps, fifteen transformers were found to supply the neighborhood. The corresponding data are given in Table 1.

**Table 1.** The various transformers of the area.

| Label | Transformer Power (kVA) | 80% (kVA) | Connected Customers | Connected Customers with Electrical Heating |
|---|---|---|---|---|
| T261 | 800 | 640 | 66 | 56 |
| T262 | 800 | 640 | 55 | 36 |
| T270 | 800 | 640 | 61 | 61 |
| T271 | 800 | 640 | 57 | 57 |
| T272 | 500 | 400 | 42 | 42 |
| T273 | 500 | 400 | 44 | 43 |
| T274 | 500 | 400 | 49 | 49 |
| T280 | 800 | 640 | 59 | 58 |
| T281 | 800 | 640 | 58 | 58 |
| T282 | 800 | 640 | 66 | 66 |
| T283 | 800 | 640 | 67 | 67 |
| T288 | 315 | 252 | 21 | 20 |
| T289 | 800 | 640 | 30 | 29 |
| T290 | 800 | 640 | 68 | 68 |
| T422 | 315 | 252 | 1 | 0 |

Although the transformers can be utilized at 100% capacity, Karlstads El- och Stadsnät, has set a limit of 80% utilization rate to ensure that the life expectancy of the transformers is not affected. This is a special characteristic of this utility not usually met by other system operators. The remaining 20% are used to deal with reactive power and other problems that may occur. In Figures 1–3, one can see the different power demands for the respective transformers' capacities.

This study focuses on four consecutive transformers (on the same line) to slightly limit the workload. Therefore, the four transformers T261, T262, T273, and T422 were selected since they seem to contain one of every kind of transformer capacity of those in the area. By comparing the transformers in Figures 1–3, it can be seen that they all have reasonably similar power demands.

The power curves are calculated by taking each transformer's maximum power consumption at every hour in 2017, comparing them with the coldest year that data is available for (2010), and creating a new table with the highest values for each hour. These new values are used as the basis for power demand in the simulation and are given in Table 2.

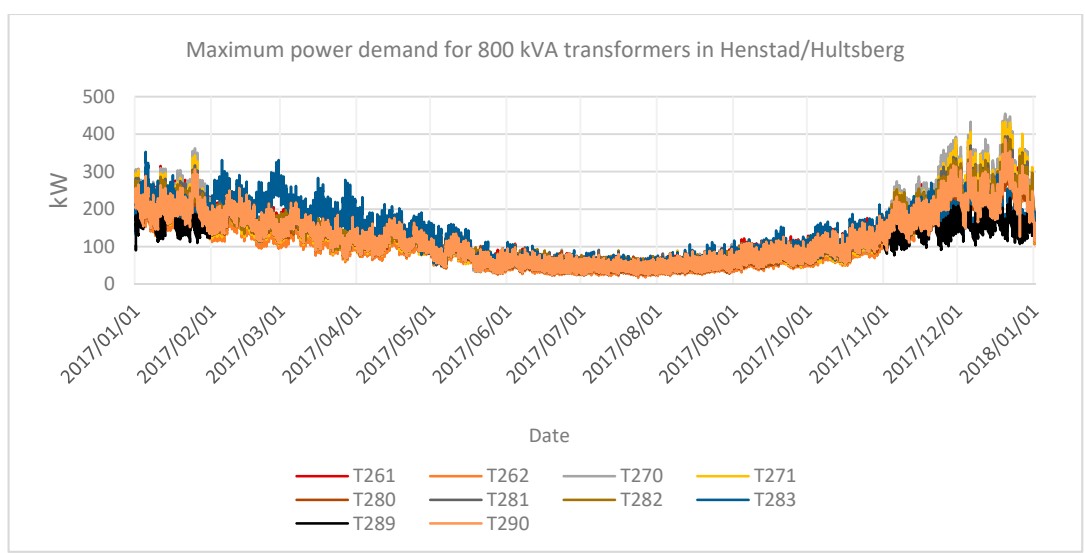

**Figure 1.** The power demand of 800 kVA transformers.

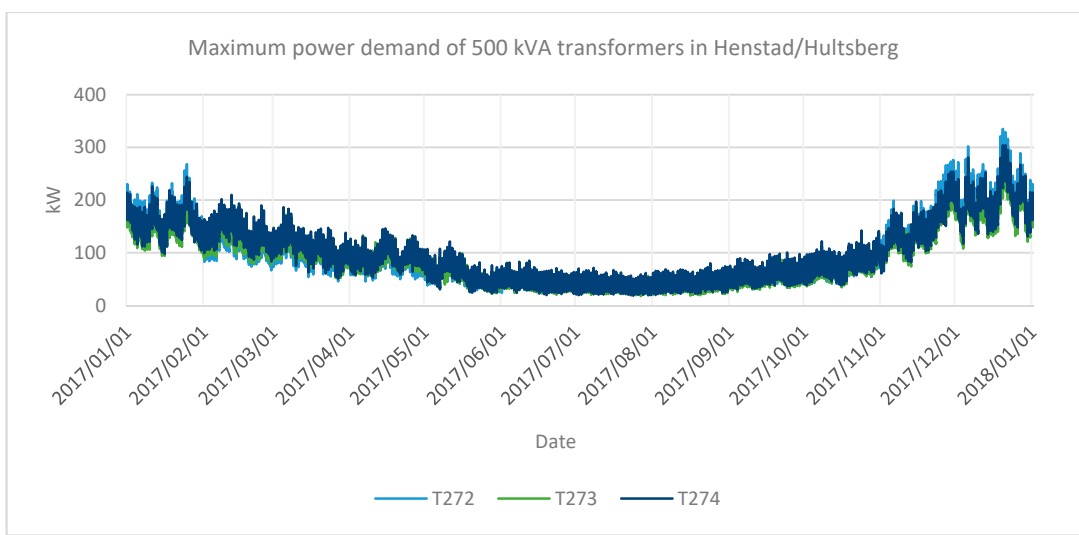

**Figure 2.** The power demand of 500 kVA transformers.

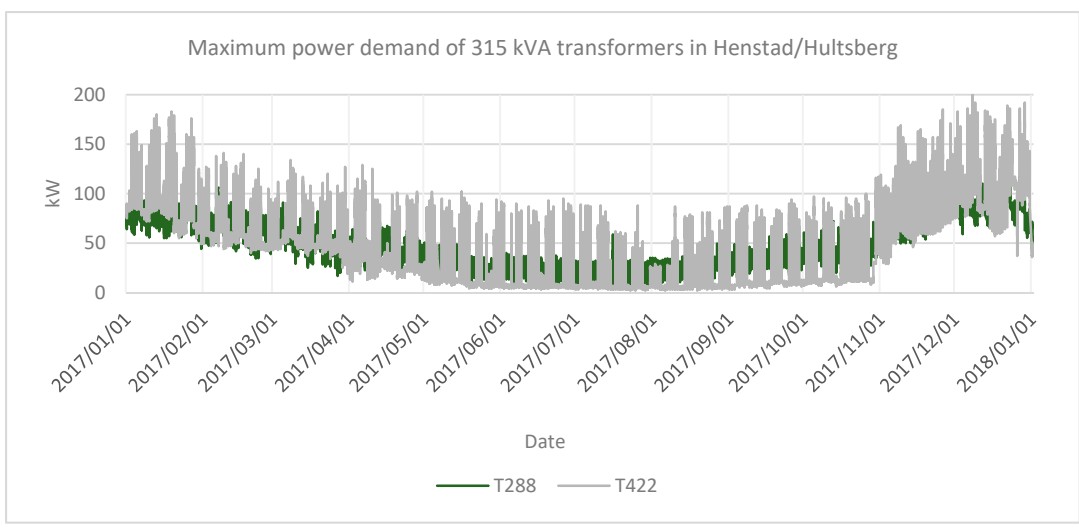

**Figure 3.** The power demand of 315 kVA transformers.

**Table 2.** Total power demand of the transformers.

| Transformer | T261 | T262 | T273 | T422 |
|---|---|---|---|---|
| Power demand GWh/year | 1.199749 | 0.964139 | 0.784951 | 0.463429 |
| Electricity demand GWh/year | 0.188939 | 0.289061 | 0.187890 | 0.463429 |
| Heating demand GWh/year | 1.010479 | 0.675078 | 0.597062 | 0.000000 |

The power demand was calculated using Equation (A1) in the Appendix A. The available energy source via the transformer could not be determined since it was considered confidential and case sensitive data and since EnergyPLAN needs such an input, that power is considered as imported, and therefore, not included in the software. However, one can add this input as a source of energy in Microsoft Excel with varying values depending on the power requirement. When calculating balance for flexible demand, Equation (A2) was used and Equation (A3) was used when other energy sources were added to the system.

The charging profile was based primarily on a constant charge throughout the day with the intention of ensuring that the worst case is covered since it is not certain that all EVs are charged at a certain time of the day. "Worst case" is when all electric vehicles are charged while the regular household demands reach their maximum values.

Thereafter, a study was also made regarding how power demands change if charging occurs only on late nights and when most of the customer activities are on the lower level (i.e., between 22:00 and 06:00). This is illustrated in Figure 4.

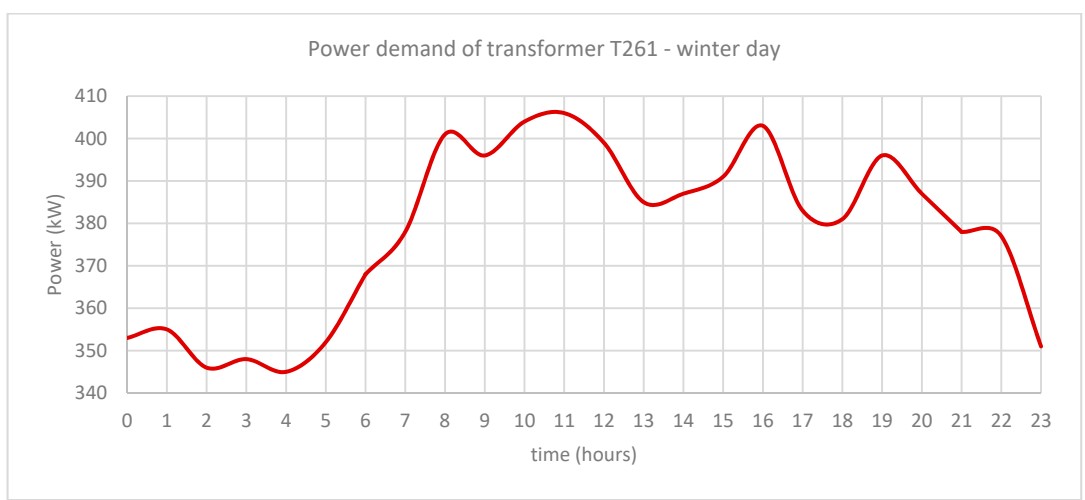

**Figure 4.** Power demand during the 24 h.

According to the above assumptions, the following charge profiles were introduced, as shown in Figure 5. For the transformers T261, T262, and T273, the "Charging households" profile was used and, for the transformer T422, the "Charging industries" profile was used. For 'Charging industries,' it is assumed that staff work between 8:00 and 16:00.

The total power demand was calculated by Equation (A4) by multiplying the percentage of the distribution file by the number of EVs and the charging capacity. In this example, Equation (A5) was used to calculate the total batteries' capacity of the EVs and Equation (A6) was used to calculate the battery storage capacity. However, storage capacity has no impact on this study and has an assumed flat-rate value of approximately 80 kWh per car. There were also tests with different amounts of EVs, both by changing the number of simultaneously charging cars and different amounts of cars available in the area.

At this point, investigations of what happens when solar cells are added to the network is made. Since the maximum permissible power of the transformer cannot exceed its peak power, the decision

was made to limit the power of the solar cells by merely trying to cover the heating demand in the respective residence. This simulates that the input electricity from the solar cells only supplies the house with electricity and is not supplied into the power network. The solar panel's household heating supply was calculated by using Equation (A9). The data in Figure 6 shows the peak solar hours in Karlstad collected from the database STRÅNG [19] and the resulting balance was calculated using Equation (A8).

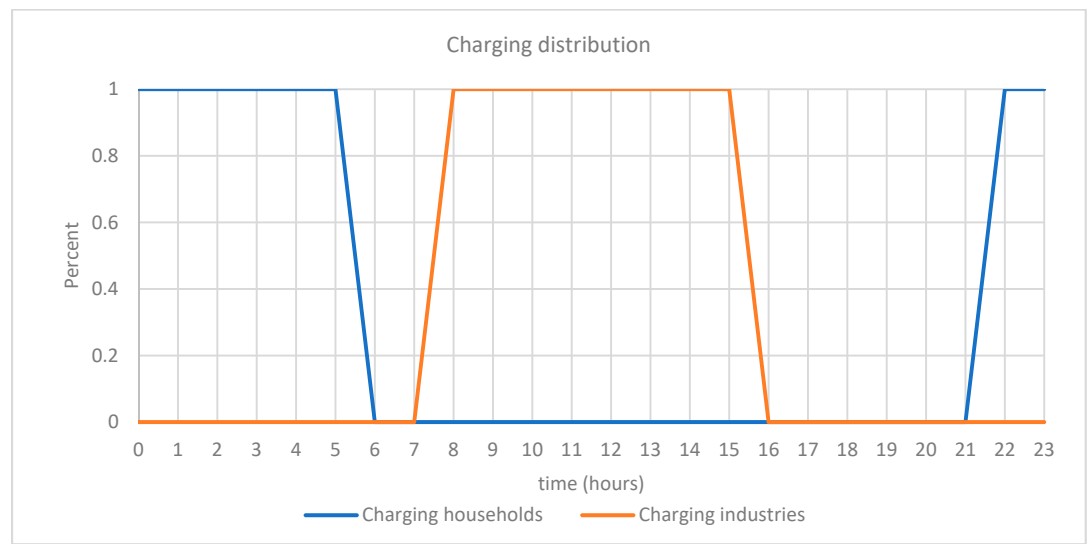

**Figure 5.** Assumed profiles.

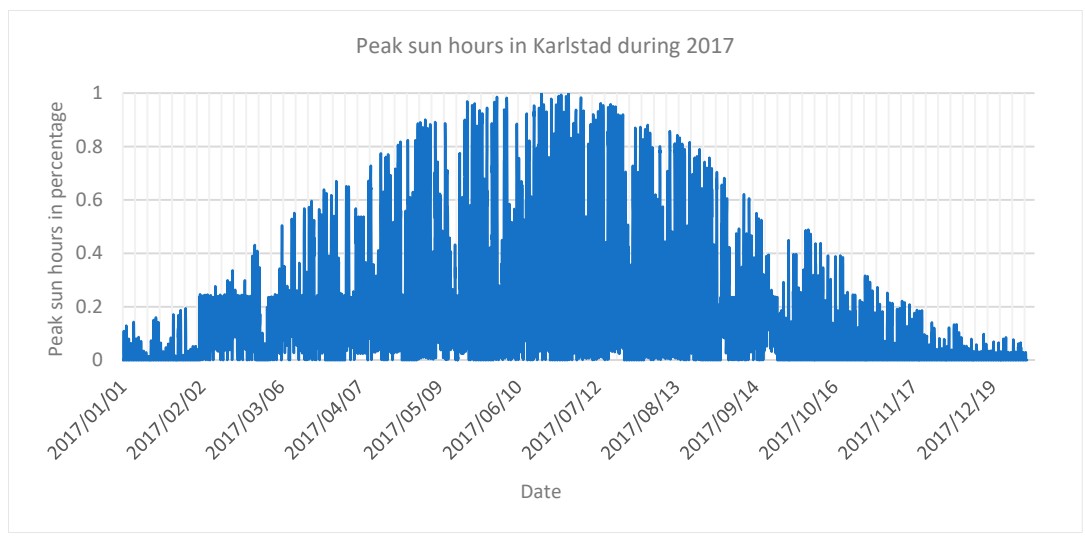

**Figure 6.** Peak sun hours in Karlstad during 2017 [19].

In order to use more of the available electricity during EVs charging, the electric heating was completely replaced by district heating to investigate whether it was possible to reduce the overall power demand via the transformer. The total energy supply was calculated, according to Equation (A10), and the balance was then calculated according to Equation (A8).

## 3. Results

In order to investigate several strategies toward a 100% RES, under high penetration of EVs, and in the presence of different sources such as PVs, district heating, and ESD, four future scenarios of grid investments have been conducted. The parameters that configure the scenarios are: (i) the number

of grid connected EVs, (ii) the level of charging rate, (iii) the charging profile, (iv) the flexible demand, and (v) the district heating versus heating using electrical units. The four scenarios are presented next.

The number of EVs was determined by using Karlstad Municipality's parking standard [20]. According to the parking standard, the number of cars per transformer are for T261 and T262 about 80 cars, T273 about 60 cars, and for T422, which is located at an industrial area, with about 40 cars. To cover more potential EVs for future needs, calculations are also made at ± 20 cars per transformer.

Regarding EVs charging, several solutions could be applied, according to ABB [21]. ABB's "Mode 3 type 2" charging box [21], charging sizes can range from 2.3 kW (equivalent to single phase 230 V and 10 A fuse) up to possibly about 22 kW at home charging (corresponding to three-phase 400 V and 32 A fuse). The latter is, however, much less common. Three-phase charging maximizes charging power without the need to upgrade the main fuse and also allows for the distribution of the load on the phases. The most common charging sizes in Sweden are currently single-phase 2.3 kW and 3.7 kW, and, sometimes, three-phase 6.9 kW and 11 kW in residential areas, since these only require a fuse of 10 and 16 A, respectively. Since upgrading to an even bigger fuse is expensive and is not always desirable, most of the electric vehicles are charging at night when there is no need for faster charging.

Electric vehicle demands were added as either 'Dump Charge' or 'Smart Charge.' Dump Charge means that charging time is defined by the user through a distribution file and that the EV is constantly charged during that time. Smart Charge means that the EVs either are not charged constantly (partial charging) or that the EVs will save electricity by using part of the electricity stored in their batteries, which can then be used to reduce the surplus of electricity produced from renewable energy sources. Thus, the amount of electricity produced can be reduced, so the overload of the grid is avoided [22].

### 3.1. Scenario 1 Parameters: Number of Electric Vehicles—Charging Rate—Charging Profile

According to the first scenario, the impact of the integration of EVs is investigated with the following parameters analyzed: the number of grid-connected vehicles, the level of charging rate, and the charging profile. The maximum power differences seen in Tables 3 and 4 are the maximum power consumption after available power have been subtracted via the transformers.

**Table 3.** Maximum power balance 3.7, 6.9, or 11 kW power—constant charging.

| T261 | Max Difference | T262 | Max Difference |
|---|---|---|---|
| Balance excluding EVs | 0 kW | Balance excluding EVs | 0 kW |
| Balance 60–3.7 | −30 kW | Balance 60—3.7 | 0 kW |
| Balance 60—6.9 | −222 kW | Balance 60—6.9 | 0 kW |
| Balance 60—11 | −468 kW | Balance 60—11 | −104 kW |
| Balance 80—3.7 | −104 kW | Balance 80—3.7 | 0 kW |
| Balance 80—6.9 | −360 kW | Balance 80—6.9 | 0 kW |
| Balance 80—11 | −688 kW | Balance 80—11 | −324 kW |
| Balance 100—3.7 | −178 kW | Balance 100—3.7 | 0 kW |
| Balance 100—6.9 | −498 kW | Balance 100—6.9 | −134 kW |
| Balance 100—11 | −908 kW | Balance 100—11 | −544 kW |
| **T273** | **Max Difference** | **T422** | **Max Difference** |
| Balance excluding EVs | 0 kW | Balance excluding EVs | 0 kW |
| Balance 40—3.7 | −35 kW | Balance 20—3.7 | −22 kW |
| Balance 40—6.9 | −163 kW | Balance 20—6.9 | −86 kW |
| Balance 40—11 | −327 kW | Balance 20—11 | −168 kW |
| Balance 60—3.7 | −109 kW | Balance 40—3.7 | −96 kW |
| Balance 60—6.9 | −301 kW | Balance 40—6.9 | −224 kW |
| Balance 60—11 | −547 kW | Balance 40—11 | −388 kW |
| Balance 80—3.7 | −183 kW | Balance 60—3.7 | −170 kW |
| Balance 80—6.9 | −439 kW | Balance 60—6.9 | −362 kW |
| Balance 80—11 | −767 kW | Balance 60—11 | −608 kW |

**Table 4.** Maximum power balance 3.7, 6.9, or 11 kW night-time charging.

| T261 | Max Difference | T262 | Max Difference |
|---|---|---|---|
| Balance excluding EVs | 0 kW | Balance excluding EVs | 0 kW |
| Balance 60—3.7 | 0 kW | Balance 60—3.7 | 0 kW |
| Balance 60—6.9 | −163 kW | Balance 60—6.9 | −60 kW |
| Balance 60—11 | −402 kW | Balance 60—11 | −303 kW |
| Balance 80—3.7 | −55 kW | Balance 80—3.7 | 0 kW |
| Balance 80—6.9 | −299 kW | Balance 80—6.9 | −197 kW |
| Balance 80—11 | −612 kW | Balance 80—11 | −518 kW |
| Balance 100—3.7 | −125 kW | Balance 100—3.7 | −22 kW |
| Balance 100—6.9 | −424 kW | Balance 100—6.9 | −98 kW |
| Balance 100—11 | −822 kW | Balance 100—11 | −509 kW |
| **T273** | **Max Difference** | **T422** | **Max Difference** |
| Balance excluding EVs | 0 kW | Balance excluding EVs | 0 kW |
| Balance 40—3.7 | 0 kW | Balance 20—3.7 | −37 kW |
| Balance 40—6.9 | −119 kW | Balance 20—6.9 | −186 kW |
| Balance 40—11 | −279 kW | Balance 20—11 | −213 kW |
| Balance 60—3.7 | −66 kW | Balance 40—3.7 | −126 kW |
| Balance 60—6.9 | −250 kW | Balance 40—6.9 | −281 kW |
| Balance 60—11 | −494 kW | Balance 40—11 | −488 kW |
| Balance 80—3.7 | −138 kW | Balance 60—3.7 | −216 kW |
| Balance 80—6.9 | −388 kW | Balance 60—6.9 | −454 kW |
| Balance 80—11 | −709 kW | Balance 60—11 | −771 kW |

Table 3 presents the maximum power balance with respect to the number of EVs under power—constant charging of 3.7, 6.9, and 11 kW. Table 4 shows how the power demands change when charging occurs only at night and presents the maximum power balance with respect to the number of EVs under night-time charging of 3.7, 6.9, and 11 kW. Most of the transformers can handle a smaller number of one-phase charged vehicles on a 16 A fuse. However, when customers want to have faster charging, the situation becomes complex because the maximum power difference increases. In particular, in cases where maximum penetration of EVs is inserted in combination with 6.9 and 11 kW charging rates, it seems unrealistic to expect a stable RES because the power balance is comparable to the transformer ratings. As such, either the transformers need to be replaced or solutions that are more flexible need to be examined.

*3.2. Scenario 2 Parameters: Number of Electric Vehicles—Charging Rate—Flexible Demand*

In the second scenario, the impact of different levels of flexible demand is investigated in relation to the number of grid-connected EVs and the charging rate. The effect of flexible demand on transformers T261, T262, T273, and T422 are shown in the Tables 5–8, respectively. In Table 4, the case T261 with 60 cars at 6.9 kW presents a maximum difference of 163 kW while, in Table 5, the same case T261 with 60 cars at 6.9 kW having flexible demand 50% presents a maximum difference of 50 kW. In Table 4, the case T273 with 60 cars at 3.7 kW presents a maximum difference of 66 kW while, in Table 7, the same case T273 with 60 cars at 3.7 kW and with flexible demand of 50% presents a zero-maximum difference. Similar conclusions can be derived by controlling all cases where, in some cases, a lot of flexibility is needed to notice any effect. The results demonstrate that flexible demand can substantially reduce power consumption.

**Table 5.** Effect of flexible demand for Transformer T261.

| T261 | Max Difference | Max 10% | Max 20% | Max 30% | Max 50% |
|---|---|---|---|---|---|
| Balance excluding EVs | 0 kW | 0 kW | 0 kW | 0 kW | 0 kW |
| Balance 60—3.7 | 0 kW | 0 kW | 0 kW | 0 kW | 0 kW |
| Balance 60—6.9 | −186 kW | −159 kW | −134 kW | −106 kW | −50 kW |
| Balance 60—11 | −432 kW | −366 kW | −300 kW | −234 kW | −126 kW |
| Balance 80—3.7 | −68 kW | −48 kW | −32 kW | −11 kW | 0 kW |
| Balance 80—6.9 | −324 kW | −269 kW | −214 kW | −158 kW | −65 kW |
| Balance 80—11 | −652 kW | −572 kW | −484 kW | −396 kW | −265 kW |
| Balance 100—3.7 | −142 kW | −113 kW | −76 kW | −39 kW | 0 kW |
| Balance 100—6.9 | −462 kW | −393 kW | −324 kW | −255 kW | −144 kW |
| Balance 100—11 T261 | −871 kW | −777 kW | −667 kW | −557 kW | −403 kW |

**Table 6.** Effect of flexible demand for Transformer T262.

| T262 | Max Difference | Max 10% | Max 20% | Max 30% | Max 50% |
|---|---|---|---|---|---|
| Balance excluding EVs | 0 kW | 0 kW | 0 kW | 0 kW | 0 kW |
| Balance 60—3.7 | 0 kW | 0 kW | 0 kW | 0 kW | 0 kW |
| Balance 60—6.9 | −83 kW | −55 kW | −24 kW | 0 kW | 0 kW |
| Balance 60—11 | −329 kW | −284 kW | −237 kW | −193 kW | −105 kW |
| Balance 80—3.7 | 0 kW | 0 kW | 0 kW | 0 kW | 0 kW |
| Balance 80—6.9 | −221 kW | −183 kW | −144 kW | −107 kW | −33 kW |
| Balance 80—11 | −549 kW | −489 kW | −428 kW | −369 kW | −252 kW |
| Balance 100—3.7 | −39 kW | −13 kW | 0 kW | 0 kW | 0 kW |
| Balance 100—6.9 | −359 kW | −312 kW | −263 kW | −217 kW | −125 kW |
| Balance 100—11 | −768 kW | −694 kW | −617 kW | −544 kW | −397 kW |

**Table 7.** Effect of flexible demand for Transformer T273.

| T273 | Max Difference | Max 10% | Max 20% | Max 30% | Max 50% |
|---|---|---|---|---|---|
| Balance excluding EVs | 0 kW | 0 kW | 0 kW | 0 kW | 0 kW |
| Balance 40—3.7 | 0 kW | 0 kW | 0 kW | 0 kW | 0 kW |
| Balance 40—6.9 | −125 kW | −108 kW | −89 kW | −70 kW | −34 kW |
| Balance 40—11 | −289 kW | −262 kW | −231 kW | −202 kW | −143 kW |
| Balance 60—3.7 | −71 kW | −59 kW | −42 kW | −27 kW | 0 kW |
| Balance 60—6.9 | −263 kW | −238 kW | −208 kW | −181 kW | −126 kW |
| Balance 60—11 | −509 kW | −467 kW | −421 kW | −378 kW | −290 kW |
| Balance 80—3.7 | −145 kW | −127 kW | −106 kW | −86 kW | −47 kW |
| Balance 80—6.9 | −401 kW | −367 kW | −328 kW | −291 kW | −218 kW |
| Balance 80—11 | −729 kW | −673 kW | −612 kW | −554 kW | −436 kW |

**Table 8.** Effect of flexible demand for Transformer T422.

| T422 | Max Difference | Max 10% | Max 20% | Max 30% | Max 50% |
|---|---|---|---|---|---|
| Balance excluding EVs | 0 kW | 0 kW | 0 kW | 0 kW | 0 kW |
| Balance 20—3.7 | −22 kW | −15 kW | −7 kW | 0 kW | 0 kW |
| Balance 20—6.9 | −86 kW | −72 kW | −58 kW | −45 kW | −17 kW |
| Balance 20—11 | −168 kW | −146 kW | −124 kW | −102 kW | −58 kW |
| Balance 40—3.7 | −96 kW | −81 kW | −66 kW | −52 kW | −22 kW |
| Balance 40—6.9 | −224 kW | −196 kW | −169 kW | −141 kW | −86 kW |
| Balance 40—11 | −419 kW | −344 kW | −300 kW | −256 kW | −168 kW |
| Balance 60—3.7 | −170 kW | −148 kW | −126 kW | −103 kW | −59 kW |
| Balance 60—6.9 | −393 kW | −321 kW | −279 kW | −238 kW | −155 kW |
| Balance 60—11 | −639 kW | −542 kW | −476 kW | −410 kW | −299 kW |

*3.3. Scenario 3 Parameters: Number of Electric Vehicles—Charging Rate—Charging Profile—District Heating*

The third scenario presents the power balance when space-heating using electrical units is gradually replaced by district heating with involved parameters acting as the number of connected EVs, the charging rate, and the charging profile. The effect of having district heating on transformers T261, T262, and T273 are provided in Tables 9–14, with both a constant charge and charging at night, respectively.

**Table 9.** Replacement of electric heating by district heating for transformer T261—constant charging.

| T261 | Max Difference | Max 90% | Max 80% | Max 70% | Max 60% |
|---|---|---|---|---|---|
| Balance excluding EVs | 0 kW | 0 kW | 0 kW | 0 kW | 0 kW |
| Balance 60—3.7 | 0 kW | 0 kW | 0 kW | 0 kW | 0 kW |
| Balance 60—6.9 | 0 kW | 0 kW | 0 kW | 0 kW | 0 kW |
| Balance 60—11 | −75 kW | −9 kW | 0 kW | 0 kW | 0 kW |
| Balance 80—3.7 | 0 kW | 0 kW | 0 kW | 0 kW | 0 kW |
| Balance 80—6.9 | 0 kW | 0 kW | 0 kW | 0 kW | 0 kW |
| Balance 80—11 | −295 kW | −207 kW | −119 kW | −31 kW | 0 kW |
| Balance 100—3.7 | 0 kW | 0 kW | 0 kW | 0 kW | 0 kW |
| Balance 100—6.9 | −99 kW | −30 kW | 0 kW | 0 kW | 0 kW |
| Balance 100—11 | −515 kW | −405 kW | −295 kW | −185 kW | −75 kW |

**Table 10.** Replacement of electric heating by district heating for transformer T261—late night charging.

| T261 | Max Difference | Max 90% | Max 80% | Max 70% | Max 60% |
|---|---|---|---|---|---|
| Balance excluding EVs | 0 kW | 0 kW | 0 kW | 0 kW | 0 kW |
| Balance 60—3.7 | 0 kW | 0 kW | 0 kW | 0 kW | 0 kW |
| Balance 60—6.9 | 0 kW | 0 kW | 0 kW | 0 kW | 0 kW |
| Balance 60—11 | −63 kW | 0 kW | 0 kW | 0 kW | 0 kW |
| Balance 80—3.7 | 0 kW | 0 kW | 0 kW | 0 kW | 0 kW |
| Balance 80—6.9 | 0 kW | 0 kW | 0 kW | 0 kW | 0 kW |
| Balance 80—11 | −283 kW | −195 kW | −107 kW | −4 kW | 0 kW |
| Balance 100—3.7 | 0 kW | 0 kW | 0 kW | 0 kW | 0 kW |
| Balance 100—6.9 | −93 kW | −24 kW | 0 kW | 0 kW | 0 kW |
| Balance 100—11 | −502 kW | −392 kW | −282 kW | −172 kW | −62 kW |

**Table 11.** Replacement of electric heating by district heating for transformer T262—constant charging.

| T262 | Max Difference | Max 90% | Max 80% | Max 70% | Max 60% |
|---|---|---|---|---|---|
| Balance excluding EVs | 0 kW | 0 kW | 0 kW | 0 kW | 0 kW |
| Balance 60—3.7 | 0 kW | 0 kW | 0 kW | 0 kW | 0 kW |
| Balance 60—6.9 | 0 kW | 0 kW | 0 kW | 0 kW | 0 kW |
| Balance 60—11 | −104 kW | −38 kW | 0 kW | 0 kW | 0 kW |
| Balance 80—3.7 | 0 kW | 0 kW | 0 kW | 0 kW | 0 kW |
| Balance 80—6.9 | 0 kW | 0 kW | 0 kW | 0 kW | 0 kW |
| Balance 80—11 | −324 kW | −236 kW | −148 kW | −60 kW | 0 kW |
| Balance 100—3.7 | 0 kW | 0 kW | 0 kW | 0 kW | 0 kW |
| Balance 100—6.9 | −134 kW | −65 kW | 0 kW | 0 kW | 0 kW |
| Balance 100—11 | −544 kW | −434 kW | −324 kW | −214 kW | −104 kW |

The case of T262 with 80 cars at 11 kW charging rate presents a power difference of 518 kW in Table 4. When 50% flexible demand is inserted, the power difference drops down to 252 kW in Table 6 while, in the case of 60% of space-heating, it is based on electricity and the 40% is covered by district heating. The power difference is already zero even with constant charging in Table 11. The results clearly indicate that the power balance is successfully achieved in most of the cases. However, maximum EVs penetration per transformer having the highest rate of charge of 11 kW is not

easily manageable because results have a high power unbalance. Overall, the late-night charging leads to better power balance.

**Table 12.** Replacement of electric heating by district heating for transformer T262—late night charging.

| T262 | Max Difference | Max 90% | Max 80% | Max 70% | Max 60% |
|---|---|---|---|---|---|
| Balance excluding EVs | 0 kW | 0 kW | 0 kW | 0 kW | 0 kW |
| Balance 60—3.7 | 0 kW | 0 kW | 0 kW | 0 kW | 0 kW |
| Balance 60—6.9 | 0 kW | 0 kW | 0 kW | 0 kW | 0 kW |
| Balance 60—11 | −89 kW | −23 kW | 0 kW | 0 kW | 0 kW |
| Balance 80—3.7 | 0 kW | 0 kW | 0 kW | 0 kW | 0 kW |
| Balance 80—6.9 | 0 kW | 0 kW | 0 kW | 0 kW | 0 kW |
| Balance 80—11 | −309 kW | −221 kW | −133 kW | −45 kW | 0 kW |
| Balance 100—3.7 | 0 kW | 0 kW | 0 kW | 0 kW | 0 kW |
| Balance 100—6.9 | −119 kW | −50 kW | 0 kW | 0 kW | 0 kW |
| Balance 100—11 | −528 kW | −418 kW | −308 kW | −198 kW | −88 kW |

**Table 13.** Replacement of electric heating by district heating for transformer T273—constant charging.

| T273 | Max Difference | Max 90% | Max 80% | Max 70% | Max 60% |
|---|---|---|---|---|---|
| Balance excluding EVs | 0 kW | 0 kW | 0 kW | 0 kW | 0 kW |
| Balance 40—3.7 | 0 kW | 0 kW | 0 kW | 0 kW | 0 kW |
| Balance 40—6.9 | 0 kW | 0 kW | 0 kW | 0 kW | 0 kW |
| Balance 40—11 | −94 kW | −50 kW | −6 kW | 0 kW | 0 kW |
| Balance 60—3.7 | 0 kW | 0 kW | 0 kW | 0 kW | 0 kW |
| Balance 60—6.9 | −68 kW | −27 kW | 0 kW | 0 kW | 0 kW |
| Balance 60—11 | −314 kW | −248 kW | −182 kW | −116 kW | −50 kW |
| Balance 80—3.7 | 0 kW | 0 kW | 0 kW | 0 kW | 0 kW |
| Balance 80—6.9 | −206 kW | −151 kW | −96 kW | −40 kW | 0 kW |
| Balance 80—11 | −534 kW | −446 kW | −358 kW | −270 kW | −182 kW |

**Table 14.** Replacement of electric heating by district heating for transformer T273—late night charging.

| T273 | Max Difference | Max 90% | Max 80% | Max 70% | Max 60% |
|---|---|---|---|---|---|
| Balance excluding EVs | 0 kW | 0 kW | 0 kW | 0 kW | 0 kW |
| Balance 40—3.7 | 0 kW | 0 kW | 0 kW | 0 kW | 0 kW |
| Balance 40—6.9 | 0 kW | 0 kW | 0 kW | 0 kW | 0 kW |
| Balance 40—11 | −85 kW | −41 kW | 0 kW | 0 kW | 0 kW |
| Balance 60—3.7 | 0 kW | 0 kW | 0 kW | 0 kW | 0 kW |
| Balance 60—6.9 | −59 kW | −18 kW | 0 kW | 0 kW | 0 kW |
| Balance 60—11 | −305 kW | −239 kW | −173 kW | −107 kW | −41 kW |
| Balance 80—3.7 | 0 kW | 0 kW | 0 kW | 0 kW | 0 kW |
| Balance 80—6.9 | −197 kW | −142 kW | −87 kW | −31 kW | 0 kW |
| Balance 80—11 | −525 kW | −437 kW | −349 kW | −261 kW | −173 kW |

*3.4. Scenario 4 Parameters: Number of Electric Vehicles—Charging Rate—Flexible Demand—District Heating*

In the fourth scenario, district heating replaces space-heating using electrical units, several levels of flexible demand are involved with parameters of the number of connected EVs and the charging rate. The effect of having several levels of flexible demand in the presence of district heating on transformers T261, T262, and T273 are provided in Tables 15–17, with charging at night, respectively. For example, the case of T262 with 80 cars at 11 kW charging rate and 50% flexibility in Table 16 presents a power difference of 3 kW. By comparing to the case T262 with 80 cars at an 11 kW charging rate and when 60% of space-heating is based on electricity and the 40% is covered by district heating in Table 12, which presents a power difference of zero, one concludes that district heating could be more important than flexible demand. Similar conclusions reveal, by comparing the other cases and, hence, district

heating in combination with flexible demand is considered as one of the most promising scenarios in this case study.

**Table 15.** Replacement of electric heating by district heating in the presence of flexible demand for transformer T261—Charging at night.

| T261 | Max Difference | Max 10% | Max 20% | Max 30% | Max 50% |
|---|---|---|---|---|---|
| Balance excluding EVs | 0 kW | 0 kW | 0 kW | 0 kW | 0 kW |
| Balance 60—3.7 | 0 kW | 0 kW | 0 kW | 0 kW | 0 kW |
| Balance 60—6.9 | 0 kW | 0 kW | 0 kW | 0 kW | 0 kW |
| Balance 60—11 | −63 kW | −8 kW | 0 kW | 0 kW | 0 kW |
| Balance 80—3.7 | 0 kW | 0 kW | 0 kW | 0 kW | 0 kW |
| Balance 80—6.9 | 0 kW | 0 kW | 0 kW | 0 kW | 0 kW |
| Balance 80—11 | −283 kW | −213 kW | −151 kW | −91 kW | 0 kW |
| Balance 100—3.7 | 0 kW | 0 kW | 0 kW | 0 kW | 0 kW |
| Balance 100—6.9 | −93 kW | −36 kW | 0 kW | 0 kW | 0 kW |
| Balance 100—11 | −502 kW | −430 kW | −357 kW | −286 kW | −139 kW |

**Table 16.** Replacement of electric heating by district heating in the presence of flexible demand for transformer T262—Charging at night.

| T262 | Max Difference | Max 10% | Max 20% | Max 30% | Max 50% |
|---|---|---|---|---|---|
| Balance excluding EVs | 0 kW | 0 kW | 0 kW | 0 kW | 0 kW |
| Balance 60—3.7 | 0 kW | 0 kW | 0 kW | 0 kW | 0 kW |
| Balance 60—6.9 | 0 kW | 0 kW | 0 kW | 0 kW | 0 kW |
| Balance 60—11 | −89 kW | −42 kW | 0 kW | 0 kW | 0 kW |
| Balance 80—3.7 | 0 kW | 0 kW | 0 kW | 0 kW | 0 kW |
| Balance 80—6.9 | 0 kW | 0 kW | 0 kW | 0 kW | 0 kW |
| Balance 80—11 | −309 kW | −247 kW | −185 kW | −120 kW | −3 kW |
| Balance 100—3.7 | 0 kW | 0 kW | 0 kW | 0 kW | 0 kW |
| Balance 100—6.9 | −119 kW | −70 kW | −20 kW | 0 kW | 0 kW |
| Balance 100—11 | −528 kW | −451 kW | −374 kW | −295 kW | −148 kW |

**Table 17.** Replacement of electric heating by district heating in the presence of flexible demand for transformer T273—Charging at night.

| T273 | Max Difference | Max 90% | Max 80% | Max 70% | Max 60% |
|---|---|---|---|---|---|
| Balance excluding EVs | 0 kW | 0 kW | 0 kW | 0 kW | 0 kW |
| Balance 40—3.7 | 0 kW | 0 kW | 0 kW | 0 kW | 0 kW |
| Balance 40—6.9 | 0 kW | 0 kW | 0 kW | 0 kW | 0 kW |
| Balance 40—11 | −85 kW | −53 kW | −22 kW | 0 kW | 0 kW |
| Balance 60—3.7 | 0 kW | 0 kW | 0 kW | 0 kW | 0 kW |
| Balance 60—6.9 | −59 kW | −30 kW | 0 kW | 0 kW | 0 kW |
| Balance 60—11 | −305 kW | −259 kW | −212 kW | −164 kW | −76 kW |
| Balance 80—3.7 | 0 kW | 0 kW | 0 kW | 0 kW | 0 kW |
| Balance 80—6.9 | −197 kW | −158 kW | −119 kW | −77 kW | −4 kW |
| Balance 80—11 | −525 kW | −464 kW | −403 kW | −340 kW | −223 kW |

*3.5. Analysis of the Results*

The integration of electric vehicles to the distribution network is a complicated task and the results of this study on a real system clearly indicate that the EV owners will share an active role during the integration phase. Moreover, the customers' behavior will ensure the viability of the integration solution in each particular case. As was expected, when the customers choose to charge with higher charging rates, analogously higher power peak demands are observed. In such a case, a deterring measure is always based on higher rates per kWh. However, there are limitations either because there

are no established business models for such cases or there are limitations on the establishment of high pricing rates because electricity is considered a social necessity.

Flexible demand through power management is a promising option to reduce overall power consumption. However, it is required that the customers agree that their electrical consumption is managed in such a manner. The usage of photovoltaics cannot analogously mitigate the problem because power production takes place during daytime while residential EVs charging takes place mostly after midnight. In such a concept, the utilization of batteries or other storage methods mitigates the problem. On the contrary, the nightly charging could be seen as peak shaving and load shifting from a residential load profile perspective.

A simple and feasible alternative is based on the usage of on-grid photovoltaic systems to charge storage units during the day. Thereafter, the energy management decision tree takes action and, when it finds an effective alternative to use the stored energy, the storage units discharge. However, this concept finds difficulties in Swedish applications during winter times. Suitable business models need to be developed between the distribution operators, the real estate companies, and the individual customers.

Another alternative is based on the replacement of the electrical consumption for space heating by district heating. In this way, practically all EVs per household could be charged with optional charging capacities between 3.7 and 11 kW. Furthermore, it is up to the customers as to whether they want to invest in something that can be cheaper over the years or continue to use electrical units for space-heating. A substantial incentive is required to encourage the customers to change their space-heating, especially if almost all vehicles are going to be replaced by EVs in the future.

Lastly, there are cases where a grid upgrade is necessary. For example, regarding transformer T422, there is no other choice except to replace it, if several EVs pop up. At present, it can handle a maximum of 10–20 EVs that charge at the lowest charging rate because there is only about 50 kW electrical power available during the winter.

## 4. Conclusions

The charging levels that have been chosen to investigate were 3.7, 6.9, and 11 kW. It was assumed that single-phase charging of 2.3 kW can be handled by all transformers, so it was not included in this study. Moreover, a charging level over 11 kW is not considered necessary at residential areas and, in addition, this charging level requires larger mains installations.

Regarding the scenarios presented in this work, regarding the pattern of EVs penetration into the electrical distribution grid, one can say that more investigation and realistic scenarios need to be examined in order to define the safe strategy for an optimal solution. This is because the insertion of new technologies and customers' needs into the electrical networks is related to several technological developments, such as on batteries, on cables, on local controllers, and energy on central energy management techniques on both a secondary and tertiary control level. In addition, the solid requirement for sustainable social development will strongly affect the landscape of the future energy systems. Regarding the electrification of the transportation sector, there will likely be an extended variation in the vehicle alternatives in the coming years. Consequently, it is not very clear how this particular market will affect these decisions in the future.

The proposed solution at this particular stage would be a combination of alternatives as those proposed and analyzed in this study. One of the most promising alternatives would be to replace the space-heating based on electric units by other heating methods, which are not dependent on electricity such as district heating. Moreover, utilization of methods for flexible demand offers attractive alternative, which, in combination with other grid investments like peak shaving or distributed storage, could be very effective.

**Author Contributions:** H.M. Investigation, S.B. Formal Analysis, S.L. Writing-Review & Editing, and A.T. Supervision.

**Funding:** This research received no external funding.

**Acknowledgments:** The authors would like to acknowledge Karlstads El- och Stadsnät AB, Sweden for providing grid data and support.

**Conflicts of Interest:** The authors declare no conflict of interest.

## Abbreviations

| | |
|---|---|
| Balance | The balance between the surplus and demand of produced heat |
| CEEP | Critical excess electricity production |
| CHP | Combined heat and power from power plants |
| $C_{CHARGER}$ | The grid connection's capacity |
| $C_{V2G}$ | Total grid connection of the total electric vehicle fleet capacity per hour |
| DHP | District heat and power |
| District Heating Production | The power produced by district heating |
| District Heating Demand | The total heat demand of connected customers |
| EEP | Excess electrical production |
| EH | District heating provided by electric boilers |
| $e_{INV}$ | Electricity production (inverters) |
| Elec. Demand | The electricity demand |
| Electrolyser | Electrolyser power consumption |
| ELT | District heating from surplus heat produced by electrolysers |
| Exp | Electricity available for export |
| Flex. and Transp. | Flexible and transport demand |
| Geothermal | Geothermal power production |
| Gr. 1 | Group 1 (representing district heating without combined heat and power) |
| Gr. 2 | Group 2 (representing district heating with small combined heat and power plants) |
| Gr. 3 | Group 3 (representing district heating with large combined heat and power plants) |
| HDH3 | Heat demand in district heating group 3 |
| HP | District heating from heat pumps |
| Hydro | Electricity production via hydro power |
| Hydro Pump | The hydro pump's power demand |
| Imp | Import of electricity—the extra electricity needed to support the grid |
| $n$ | Number of electric vehicles |
| $n_{EV}$ | Number of electric vehicles |
| $n_{household}$ | Number of households |
| $n_{norm}$ | Number of vehicles per household according to Karlstad's parking norm |
| $P_{EV}$ | Total power demand when charging EV's |
| Pdemand | Total power demand per hour |
| PP | Electricity production via power plants |
| PPV/h | Power produced by solar panels per hour |
| PPVtotal | Total amount of power desired from solar panels |
| $P_{supply}$ | Total power supply per hour |
| $P_{total\ demand}$ | The total annual power demand in the grid |
| $P_{total\ supply}$ | The total annual power supply to the grid |
| $P_{transformer}$ | Available power via the transformer per hour |
| QDH3 | Heat production in district heating group 3 |
| RES | Electricity production via renewable energy sources |
| $\Sigma_{demand/h}$ | The total sum of the power demand per hour |
| $\Sigma_{hours}$ | The sum of hours in 366 days |
| Solar | District heating via solar panels |
| StabLoad | The grid stability in % |
| Storage | The amount of district heating stored in thermal storage systems |

| $S_{V2G}$ | The total storage capacity of the EV batteries |
|---|---|
| $T_{80\%}$ | 80% of the available power through a transformer |
| Turbine | Electricity production via turbines |
| $V2G_{MAX}$ | The maximum number of vehicles in traffic simultaneously in per cent |
| $V2G_{CONNECTED}$ | The maximum number of electric vehicles connected to the grid at the same time in per cent |
| Waste CSHP | District heating via waste combustion and from industries |
| $\delta_{PV}$ | Distribution of the solar panels' efficiency |
| $\delta_{V2G}$ | Distribution the power demand of EV's |
| $\mu_{INV}$ | The efficiency of the connection between the batteries and the grid (inverter) |
| $\rho_{DH3}$ | Heat losses in district heating group 3 |

## Appendix A

The total power consumption per year is given by the equation:

$$P_{total\ demand} = \frac{\sum_{power\ demand/year}}{\sum_{hours}} \tag{A1}$$

The available transformer power is:

$$P_{transformer} = IF(P_{demand} > T_{80\%}\ THEN\ P_{transformer} = T_{80\%},\ ELSE\ P_{transformer} = P_{demand}) \tag{A2}$$

and

$$P_{transformer} = \begin{array}{l} IF(P_{demand} - P_{supply} > T_{80\%},\ THEN\ P_{transformer} \\ = T_{80\%},\ ELSE\ P_{transformer} = P_{demand} - P_{supply}) \end{array} \tag{A3}$$

The power demand of electric vehicles is provided by Reference [20] while charging electric vehicles:

$$P_{EV} = n * \delta_{V2G} * C_{charger} \tag{A4}$$

Total capacity of the batteries of electric vehicles:

$$C_{V2G} = C_{charger} * V2G_{connected} * (1 - V2G_{max}) + V2G_{max} * \left(1 - \frac{\delta_{V2G}}{Max(\delta_{V2G})}\right) \tag{A5}$$

The batteries' storage capacity:

$$S_{V2G} = s_{V2G} - \frac{e_{inv}}{\mu_{inv}} \tag{A6}$$

The number of electric vehicles:

$$n_{EV} \approx n_{norm} * n_{households} \tag{A7}$$

The balance of power:

$$Balance = P_{total\ demand} - P_{total\ supply} \tag{A8}$$

The solar panels' supply to the grid by using Reference [20] and the solar panels' power supply per hour:

$$P_{PV/h} = \delta_{PV} * P_{PVtotal} \tag{A9}$$

Power supply from district heating by using Reference [20] and the power supply per hour provided by district heating:

$$H_{DH3} = Q_{DH3} * (1 - \rho_{DH3}) \tag{A10}$$

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
