# Peer review of "Electric Vehicle Penetration in Distribution Network: A Swedish Case Study"

_asi, doi:10.3390/asi2030019_

Reviewer 1 Report

This paper presented a timely topic. However, the paper is not well presented and it need a lot of work. 

- The authors need to rewrite the paper. I advise the authors to check other Journal papers in the filed.

- abstract section is too short and did not sound significant .   I advise the authors to check other Journal papers in the filed.

- The authors need to extend the literature review to include the main work on the same are. In addition, authors need to be critical about other people work and show the gaps (not only listed them).

- need to clarify and specify what is the new in this paper and the contribution?

-the methods or model is not new or contribution? Authors need clarify and specify if there is anything new

- the tools and methods that used in the paper need more clarifications for example add flowchart, what is the input and output of the model? How it works? How it will help us in the paper ?

- In data sections.  figures 2 and 3 are an extra figures give the same idea of figure 1 so you need better way to present the data? Figures are fuzzy? Clarify the size and resolution of the data?

Results section, is too poor and repeated the same figures and tables without new information or new significant conclusion. The authors need to be critical about the results (figures and tables).

Author Response

Dear reviewers, 

The authors appreciate reviewers’ productive comments, which have substantially contributed in improving this manuscript. All remarks, to a certain degree, have been covered as analysed below. 

“Response to Reviewer#1:

This paper presented a timely topic. However, the paper is not well presented, and it need a lot of work. 

Thank you for your feedback! 

- The authors need to rewrite the paper. I advise the authors to check other Journal papers in the field.

The manuscript has been rewritten to include reviewers’ comments. 

- abstract section is too short and did not sound significant.   I advise the authors to check other Journal papers in the field.

The abstract has been expanded. The following description has been added: 
“The importance of this research is mainly based on the fact that it represents a real network, as it operates today, which is analysed using the expected electric vehicle penetration. The aim is to investigate the expansion needs for maintaining the current quality for service despite the addition of new loads. Also, the regional and national special regulatory and operational requirements are taken into account and described in this work.”

- The authors need to extend the literature review to include the main work on the same are. In addition, authors need to be critical about other people work and show the gaps (not only listed them).

Bibliography has been rewritten to include recent work on the subject.

- need to clarify and specify what is the new in this paper and the contribution?

Introduction has been partially rewritten to present in a clearer manner this paper’s contribution. 

-the methods or model is not new or contribution? Authors need clarify and specify if there is anything new

This work uses a real network and its results have been applied to a real system. This has been better clarified in the text. 

- the tools and methods that used in the paper need more clarifications for example add flowchart, what is the input and output of the model? How it works? How it will help us in the paper?

The manuscript has been rewritten to improve readability. 

- In data sections  figures 2 and 3 are an extra figures give the same idea of figure 1 so you need better way to present the data? Figures are fuzzy? Clarify the size and resolution of the data?

All figures have been improved. The authors also have decided to remove partially redundant figures in the results section. 

Results section, is too poor and repeated the same figures and tables without new information or new significant conclusion. The authors need to be critical about the results (figures and tables).

Results are better explained, and all figures were redrawn. The discussions section has been merged with the results and all figures were removed. 

Response to Reviewer#2:

Comment 1:

Problem statement not clearly defined in Abstract.

The abstract is expanded as follows: 

“The importance of this research is mainly based on the fact that it represents a real network, as it operates today, which is analyzed using the expected electric vehicle penetration. The aim is to investigate the expansion needs for maintaining the current quality for service despite the addition of new loads. Also, the regional and national special regulatory and operational requirements are taken into account and described in this work.”

Comment 2:

The flow of idea in the introduction section is not regular. (One paragraph discussing one issue and next paragraph discusses a totally different issue with no link in-between)

Manuscript’s flow has been rearranged to bridge gaps improve readability. 

Comment 3:

Reference formatting in text is not correct.

Bibliography formatting has been corrected. 

Comments 4:

Which type of problem arises due to increase in penetration of EV to electrical grid? Include references also.

The following relevant references have been added: 

4.         H. Morais, T. Sousa, Z. Vale and P. Faria, "Evaluation of the electric vehicle impact in the power demand curve in a smart grid environment," Energy Conversion and Management, vol. 82, p. 268–282, 2014. 

5.         "Supporting involvement of electric vehicles in distribution grids: Lowering the barriers for a proactive integration," Energy, vol. 134, pp. 458-468, 2017. 

6.         F. Wu and R. Sioshansi, "A two-stage stochastic optimization model for scheduling electric vehicle charging loads to relieve distribution-system constraints," Transportation Research Part B, vol. 102, pp. 55-82, 2017. 

7.         J. Hu, S. You, M. Lind and J. Østergaard, "Coordinated Charging of Electric Vehicles for Congestion Prevention in the Distribution Grid," IEEE Transactions on Smart Grid, vol. 5, no. 2, pp. 703-711, 2014.

Comment 5: (Page 3 line 94)

Number of EV at Karlstads El- och Stadsnät, Henstad/Hultsberg area in Karlstad power network. 

The authors were not able to distinguish in detail the exact number and charging pattern of Electric Vehicles in this area. This is an issue to be possibly investigated in our future work. 

Comment 6: (Page 3 line 101-102, 104)

Either transformer is running at 80% or utilizing rate is 80%? What type of other problems that may occur?

The text has been rewritten as follows: 

“Although the transformers can be utilized at 100% capacity Karlstads El- och Stadsnät, has set a limit of 80% utilization rate to ensure that the life expectancy of the transformers is not affected. This is a special characteristic of this utility not usually met to other system operators.”

Comment 7:

Why only considering/assuming data? Why not a real-time/practical data?

The network as it operates today is considered in this exercise. The relevant data for the expected photovoltaic production and electric vehicle consumption has been decided, to a certain degree, from the bibliography. According to our opinion, this is a safe approach given the lack of data for the specific region. However, the expected load due to electric vehicles and PV production in the region will be investigated in our future work. 

Comment 8:

All the results have been added (dumped) into the manuscript. Need brief explanation of every result if possible.

Results are better explained; figures were redrawn and redundant figures were removed. The discussions section has been merged with the results. 

Can you please revise all the results in term of Best-case-scenario and worst-case-scenario, if possible?

This proposal could improve this manuscript’s readability, however, the current structure, to the opinion of the authors, is also adequate for this analysis. 

Response to Reviewer#3:

This paper deals with the very interesting topics of renewables and EVs. However, it is very difficult to understand the aim of the paper and there several criticisms. I try to list some of them, at least the more evident I have detected by reading the paper.

(i) it is not clear what is the real goal of this paper. The authors themselves seem to have not clear it. Actually, they state that: 1: "it aims at simulating the use of renewable energy in the form of different energy sources for high penetration of EV in the Swedish power grid". In this specific case it not clear to me how they pursuit this objective, which drivers are behind the model. Then they state: 2. "the purpose is to examine the demands in order to cope with the needs that may arise". Here, so the purpose are the demands examination? 3. "to create a better understanding of how renewable energy affects the power balance and future investments". So, in this case I would expect voltage balance and/or economic analysis on case studies, for example.

This manuscript’s abstract, introduction and conclusions have been expanded as follows to make clearer the aim and contribution of this work. 

Then, the authors should clarify if they are dealing with solar cells or PV panels. 

Small Photovoltaic installations are considered from the power system point of view; however, more attention is given to the connection of electric vehicles. To alleviate any confusion, the title has been changed to remove the renewables factor and minor modifications were done at the introduction. 

The authors, moreover, should clarify how and when they have considered battery storages. Equation 6 gives the calculation of the capacity of the battery. But what about the size, the inverter, the state of charge and all typical parameters of the battery? Which battery has been considered in the study? How was it dimensioned on the ground of the specific case study? How does it influence the results?

Indeed, these are important factors with regard the high penetration of EVs in the distribution network in general. On grid level, as this study aims, this kind of investigation is too deep and extended to be integrated in this particular manuscript. To this end, another project currently runs which addresses these particular issues. The authors have decided not to consider in such depth the storage question in this manuscript and as such, separate research activities have been allocated for the above questions. 

There is a general lack of information in the data collection. For example, 

The authors observe limitations due to the fact they use a real network, which is not completely available in an open manner. 

Line 54: has the installed power from solar panels increased by 65% from 2016 or only in the past year? The sentence is misleading.

This sentence has been corrected. Now it reads: “Photovoltaic (PV) systems become increasingly popular and only in 2017, the installed power from solar panels has increased by 65% from 2016.”

Graphs in Fig.1,2 and 3 are not clear. I suggest enlarging the scale on the y-axis in order to see the fluctuations and the trends of the different curves. Moreover, I also suggest diminishing the width of each line, since they contraposition does not permit to highlight differences among the different curves.

Graphs in Fig. 1,2 and 3 have been redrawn according to reviewer’s proposal.  

In Fig.6 how is it possible to record the peak sun hours at 00:00? Typically a PV production may be recorded from 06:00 AM to 17(18):00 PM. Please correct.

This figure has been corrected to avoid confusion and improve readability. It expresses the percentage of sun hours per day. 

What is the methodology used? What is behind the simulations? How where they conducted? 

The authors apply the load flow simulations on energyPLAN using the real network of the region of Henstad/Hultsbergin the northwest of Karlstad, Sweden. 

The mention of the energyPLAN is not sufficient to state the validity of the simulation. 

But what is more surprising is the total lack of literature review. The list of references does not mean that a critical study was made. What is the novelty of this paper in relation to the existing literature? In addition, there is a huge amount of papers dealing with the same issue (if at least the issue of the manuscript is to simulate the penetration of EVs in the power grid with the balance of energy from renewable sources).

Bibliography has been completely rewritten. 

Nomenclature is absent.

The appendix provided to better explain the procedure applied in this paper in which an “abbreviations table” is included. 

The paper is not clear, neither in the scope nor in the methodology.

The paper has been rewritten in order to improve readability and structure:

i) The abstract has been expanded. The following description has been added: 
“The importance of this research is mainly based on the fact that it represents a real network, as it operates today, which is analysed using the expected electric vehicle penetration. The aim is to investigate the expansion needs for maintaining the current quality for service despite the addition of new loads. Also, the regional and national special regulatory and operational requirements are taken into account and described in this work.”

ii) Bibliography has been rewritten to include recent work on the subject.The following relevant references have been added: 

4.         H. Morais, T. Sousa, Z. Vale and P. Faria, "Evaluation of the electric vehicle impact in the power demand curve in a smart grid environment," Energy Conversion and Management, vol. 82, p. 268–282, 2014. 

5.         "Supporting involvement of electric vehicles in distribution grids: Lowering the barriers for a proactive integration," Energy, vol. 134, pp. 458-468, 2017. 

6.         F. Wu and R. Sioshansi, "A two-stage stochastic optimization model for scheduling electric vehicle charging loads to relieve distribution-system constraints," Transportation Research Part B, vol. 102, pp. 55-82, 2017. 

7.         J. Hu, S. You, M. Lind and J. Østergaard, "Coordinated Charging of Electric Vehicles for Congestion Prevention in the Distribution Grid," IEEE Transactions on Smart Grid, vol. 5, no. 2, pp. 703-711, 2014.

iii) Introduction has been partially rewritten to present in a clearer manner this paper’s contribution. 

iv) All figures have been improved and/or the redundant figures were removed. 

 v) Results are better explained, and all figures were redrawn. The discussions section has been merged with the results. 

With the above, we hope that we were able to cover reviewers’ comments and we are looking forward for editor’s final decision. 

Sincerely yours, 

The authors

Reviewer 2 Report

Comment 1:

Problem statement not clearly defined in Abstract.

Comment 2:

The flow of idea in the introduction section is not regular. (One paragraph discussing one issue and next paragraph discusses a totally different issue with no link in-between)

Comment 3:

Reference formatting in text is not correct.

Comments 4:

Which type of problem arises due to increase in penetration of EV to electrical grip? Include references also.

Comment 5: (Page 3 line 94)

Number of EV at Karlstads El- och Stadsnät, Henstad/Hultsberg area in Karlstad power network

Comment 6: (Page 3 line 101-102, 104)

Either transformer is running at 80% or utilizing rate is 80%? What type of other problems that may occur?

Comment 7:

Why only considering/assuming data? Why not a real-time/practical data?

Comment 8:

All the results have been added (dumped) into the manuscript. Need brief explanation of every result if possible.

Can you please revise all the results in term of Best-case-scenario and worst-case-scenario, if possible?

Author Response

(The authors gave the same response as above.)

Reviewer 3 Report

This paper deals with the very interesting topics od renewables and EVs. However, it is very difficult to understand the aim of the paper and there several criticisms. I try to list some of them, at least the more evident I have detected by reading the paper.

(i) it is not clear what is the real goal of this paper. The authors themselves seem to have not clear it. Actually, they state that: 1: "it aims at simulating the use of renewable energy in the form of different energy sources for high penetration of EV in the Swedish power grid". In this specific case it not clear to me how they pursuit this objective, which drivers are behind the model. Then they state: 2. "the purpose is to examine the demands in order to cope with the needs that may arise". Here, so the purpose are the demands examination? 3. "to create a better understanding of how renewable energy affects the power balance and future investments". So, in this case I would expect voltage balance and/or economic analysis on case studies, for example.

Then, the authors should clarify if they are dealing with solar cells or PV panels. 

The authors, moreover, should clarify how and when they have considered battery storages. Equation 6 gives the calcualtion of the capacity of the battery. But what about the size, the inverter, the state of charge and all typical parameters of the battery? Which battery has been considered in the study? How was it dimensionated on the ground of the specfic case study? How does it influence the results?

There is a general lack of information in the data collection. For example, 

Line 54: has the installed power from solar panels increased by 65% from 2016 or only in the past year? The sentence is misleading.

Graphs in Fig.1,2 and 3 are not clear. I suggest enlarging the scale on the y-axis in order to see the fluctuations and the trends of the different curves. Moreover, I also suggest to diminish the width of each line, since they contraposition does not permit to highlight differences among the different curves.

In Fig.6 how is it possible to record the peak sun hours at 00:00? Typically a PV production may be recorded from 06:00 AM to 17(18):00 PM. Please correct.

What is the methodology used? What is behind the simulations? HOw where they conducted? 

The mention of the energyPLAN is not sufficient to state the validity of the simulation. 

But what is more surprising is the total lack of literature review. The list of references does not mean that a critical study was made. What is the novelty of this paper in relation to the existing literature? In addition, there is a huge amount of papers dealing with the same issue (if at least the issue of the manuscript is to simulate the penetration of EVs in the power grid with the balance of energy from renewable sources). 

Nomenclature is absent.

The paper is not clear, neither in the scope nor in the methodology. 

Author Response

Dear reviewers, 

The authors appreciate reviewers’ productive comments, which have substantially contributed in improving this manuscript. All remarks, to a certain degree, have been covered as analysed below. 

“Response to Reviewer#1:

This paper presented a timely topic. However, the paper is not well presented, and it need a lot of work. 

Thank you for your feedback! 

- The authors need to rewrite the paper. I advise the authors to check other Journal papers in the field.

The manuscript has been rewritten to include reviewers’ comments. 

- abstract section is too short and did not sound significant.   I advise the authors to check other Journal papers in the field.

The abstract has been expanded. The following description has been added: 
“The importance of this research is mainly based on the fact that it represents a real network, as it operates today, which is analysed using the expected electric vehicle penetration. The aim is to investigate the expansion needs for maintaining the current quality for service despite the addition of new loads. Also, the regional and national special regulatory and operational requirements are taken into account and described in this work.”

- The authors need to extend the literature review to include the main work on the same are. In addition, authors need to be critical about other people work and show the gaps (not only listed them).

Bibliography has been rewritten to include recent work on the subject.

- need to clarify and specify what is the new in this paper and the contribution?

Introduction has been partially rewritten to present in a clearer manner this paper’s contribution. 

-the methods or model is not new or contribution? Authors need clarify and specify if there is anything new

This work uses a real network and its results have been applied to a real system. This has been better clarified in the text. 

- the tools and methods that used in the paper need more clarifications for example add flowchart, what is the input and output of the model? How it works? How it will help us in the paper?

The manuscript has been rewritten to improve readability. 

- In data sections  figures 2 and 3 are an extra figures give the same idea of figure 1 so you need better way to present the data? Figures are fuzzy? Clarify the size and resolution of the data?

All figures have been improved. The authors also have decided to remove partially redundant figures in the results section. 

Results section, is too poor and repeated the same figures and tables without new information or new significant conclusion. The authors need to be critical about the results (figures and tables).

Results are better explained, and all figures were redrawn. The discussions section has been merged with the results and all figures were removed. 

Response to Reviewer#2:

Comment 1:

Problem statement not clearly defined in Abstract.

The abstract is expanded as follows: 

“The importance of this research is mainly based on the fact that it represents a real network, as it operates today, which is analyzed using the expected electric vehicle penetration. The aim is to investigate the expansion needs for maintaining the current quality for service despite the addition of new loads. Also, the regional and national special regulatory and operational requirements are taken into account and described in this work.”

Comment 2:

The flow of idea in the introduction section is not regular. (One paragraph discussing one issue and next paragraph discusses a totally different issue with no link in-between)

Manuscript’s flow has been rearranged to bridge gaps improve readability. 

Comment 3:

Reference formatting in text is not correct.

Bibliography formatting has been corrected. 

Comments 4:

Which type of problem arises due to increase in penetration of EV to electrical grid? Include references also.

The following relevant references have been added: 

4.         H. Morais, T. Sousa, Z. Vale and P. Faria, "Evaluation of the electric vehicle impact in the power demand curve in a smart grid environment," Energy Conversion and Management, vol. 82, p. 268–282, 2014. 

5.         "Supporting involvement of electric vehicles in distribution grids: Lowering the barriers for a proactive integration," Energy, vol. 134, pp. 458-468, 2017. 

6.         F. Wu and R. Sioshansi, "A two-stage stochastic optimization model for scheduling electric vehicle charging loads to relieve distribution-system constraints," Transportation Research Part B, vol. 102, pp. 55-82, 2017. 

7.         J. Hu, S. You, M. Lind and J. Østergaard, "Coordinated Charging of Electric Vehicles for Congestion Prevention in the Distribution Grid," IEEE Transactions on Smart Grid, vol. 5, no. 2, pp. 703-711, 2014. 

Comment 5: (Page 3 line 94)

Number of EV at Karlstads El- och Stadsnät, Henstad/Hultsberg area in Karlstad power network. 

The authors were not able to distinguish in detail the exact number and charging pattern of Electric Vehicles in this area. This is an issue to be possibly investigated in our future work. 

Comment 6: (Page 3 line 101-102, 104)

Either transformer is running at 80% or utilizing rate is 80%? What type of other problems that may occur?

The text has been rewritten as follows: 

“Although the transformers can be utilized at 100% capacity Karlstads El- och Stadsnät, has set a limit of 80% utilization rate to ensure that the life expectancy of the transformers is not affected. This is a special characteristic of this utility not usually met to other system operators.”

Comment 7:

Why only considering/assuming data? Why not a real-time/practical data?

The network as it operates today is considered in this exercise. The relevant data for the expected photovoltaic production and electric vehicle consumption has been decided, to a certain degree, from the bibliography. According to our opinion, this is a safe approach given the lack of data for the specific region. However, the expected load due to electric vehicles and PV production in the region will be investigated in our future work. 

Comment 8:

All the results have been added (dumped) into the manuscript. Need brief explanation of every result if possible.

Results are better explained; figures were redrawn and redundant figures were removed. The discussions section has been merged with the results. 

Can you please revise all the results in term of Best-case-scenario and worst-case-scenario, if possible?

This proposal could improve this manuscript’s readability, however, the current structure, to the opinion of the authors, is also adequate for this analysis. 

Response to Reviewer#3:

This paper deals with the very interesting topics of renewables and EVs. However, it is very difficult to understand the aim of the paper and there several criticisms. I try to list some of them, at least the more evident I have detected by reading the paper.

(i) it is not clear what is the real goal of this paper. The authors themselves seem to have not clear it. Actually, they state that: 1: "it aims at simulating the use of renewable energy in the form of different energy sources for high penetration of EV in the Swedish power grid". In this specific case it not clear to me how they pursuit this objective, which drivers are behind the model. Then they state: 2. "the purpose is to examine the demands in order to cope with the needs that may arise". Here, so the purpose are the demands examination? 3. "to create a better understanding of how renewable energy affects the power balance and future investments". So, in this case I would expect voltage balance and/or economic analysis on case studies, for example.

This manuscript’s abstract, introduction and conclusions have been expanded as follows to make clearer the aim and contribution of this work. 

Then, the authors should clarify if they are dealing with solar cells or PV panels. 

Small Photovoltaic installations are considered from the power system point of view; however, more attention is given to the connection of electric vehicles. To alleviate any confusion, the title has been changed to remove the renewables factor and minor modifications were done at the introduction. 

The authors, moreover, should clarify how and when they have considered battery storages. Equation 6 gives the calculation of the capacity of the battery. But what about the size, the inverter, the state of charge and all typical parameters of the battery? Which battery has been considered in the study? How was it dimensioned on the ground of the specific case study? How does it influence the results?

Indeed, these are important factors with regard the high penetration of EVs in the distribution network in general. On grid level, as this study aims, this kind of investigation is too deep and extended to be integrated in this particular manuscript. To this end, another project currently runs which addresses these particular issues. The authors have decided not to consider in such depth the storage question in this manuscript and as such, separate research activities have been allocated for the above questions. 

There is a general lack of information in the data collection. For example, 

The authors observe limitations due to the fact they use a real network, which is not completely available in an open manner. 

Line 54: has the installed power from solar panels increased by 65% from 2016 or only in the past year? The sentence is misleading.

This sentence has been corrected. Now it reads: “Photovoltaic (PV) systems become increasingly popular and only in 2017, the installed power from solar panels has increased by 65% from 2016.”

Graphs in Fig.1,2 and 3 are not clear. I suggest enlarging the scale on the y-axis in order to see the fluctuations and the trends of the different curves. Moreover, I also suggest diminishing the width of each line, since they contraposition does not permit to highlight differences among the different curves.

Graphs in Fig. 1,2 and 3 have been redrawn according to reviewer’s proposal.  

In Fig.6 how is it possible to record the peak sun hours at 00:00? Typically a PV production may be recorded from 06:00 AM to 17(18):00 PM. Please correct.

This figure has been corrected to avoid confusion and improve readability. It expresses the percentage of sun hours per day. 

What is the methodology used? What is behind the simulations? How where they conducted? 

The authors apply the load flow simulations on energyPLAN using the real network of the region of Henstad/Hultsbergin the northwest of Karlstad, Sweden. 

The mention of the energyPLAN is not sufficient to state the validity of the simulation. 

But what is more surprising is the total lack of literature review. The list of references does not mean that a critical study was made. What is the novelty of this paper in relation to the existing literature? In addition, there is a huge amount of papers dealing with the same issue (if at least the issue of the manuscript is to simulate the penetration of EVs in the power grid with the balance of energy from renewable sources).

Bibliography has been completely rewritten. 

Nomenclature is absent.

The appendix provided to better explain the procedure applied in this paper in which an “abbreviations table” is included. 

The paper is not clear, neither in the scope nor in the methodology.

The paper has been rewritten in order to improve readability and structure:

i) The abstract has been expanded. The following description has been added: 
“The importance of this research is mainly based on the fact that it represents a real network, as it operates today, which is analysed using the expected electric vehicle penetration. The aim is to investigate the expansion needs for maintaining the current quality for service despite the addition of new loads. Also, the regional and national special regulatory and operational requirements are taken into account and described in this work.”

ii) Bibliography has been rewritten to include recent work on the subject.The following relevant references have been added: 

4.         H. Morais, T. Sousa, Z. Vale and P. Faria, "Evaluation of the electric vehicle impact in the power demand curve in a smart grid environment," Energy Conversion and Management, vol. 82, p. 268–282, 2014. 

5.         "Supporting involvement of electric vehicles in distribution grids: Lowering the barriers for a proactive integration," Energy, vol. 134, pp. 458-468, 2017. 

6.         F. Wu and R. Sioshansi, "A two-stage stochastic optimization model for scheduling electric vehicle charging loads to relieve distribution-system constraints," Transportation Research Part B, vol. 102, pp. 55-82, 2017. 

7.         J. Hu, S. You, M. Lind and J. Østergaard, "Coordinated Charging of Electric Vehicles for Congestion Prevention in the Distribution Grid," IEEE Transactions on Smart Grid, vol. 5, no. 2, pp. 703-711, 2014.

iii) Introduction has been partially rewritten to present in a clearer manner this paper’s contribution. 

iv) All figures have been improved and/or the redundant figures were removed. 

 v) Results are better explained, and all figures were redrawn. The discussions section has been merged with the results. 

With the above, we hope that we were able to cover reviewers’ comments and we are looking forward for editor’s final decision. 

Sincerely yours, 

The authors

Round  2

Reviewer 1 Report

- The  authors need to be critical about other people work by showing the gaps and limitation (not only listed them).

- need to clarify and specify what is the new in this paper and the contribution? is not enough to say is a real system (there is a lot of paper based on real system and data)

- still the  method that used in the paper need more clarifications by adding flowchart and explain  what is the input and output of the model? How it works? How it will help us in the paper?

- Clarify the size and resolution of the data? as the authors used real system what is the size the data that used and how they divide the data?

- Results section, is still poor and repeated the same figures and tables without new information or new significant conclusion. The authors need to be critical about the results (figures and tables) and move the analysis close to each table or figure.

Author Response

Dear editor, 

The authors appreciate reviewers’ second round productive comments, which have to a certain degree contributed in improving this manuscript. In some cases, the comments were not so clear to the authors. However, corrections and updated have been done. We would also like to mention that some comments were not compatible to the manuscript content is specific parts.  

We are looking forward for editor’s final decision. 

Sincerely yours, 

The authors

“Response to Reviewer#1:

- The  authors need to be critical about other people work by showing the gaps and limitation (not only listed them).

Response: The introduction has been rewritten in order to establish the background of the submitted research topic as well as to present similar work in the literature. However, as in all case studies, this case study is unique because is based on the individual characteristics that the under investigation electrical grid brings. In addition, the grid investments and thus, the future scenarios are meant to be compared to the present status of the grid, a comparison that appears in the results.

- need to clarify and specify what is the new in this paper and the contribution? is not enough to say is a real system (there is a lot of paper based on real system and data)

Response: The “new” that the paper presents is located to the extended analysis of the high penetration of EVs in a typical Swedish distribution grid, which is investigated on the basis of realistic future scenarios by using a proven scientific tool for large-scale grids planning. The “contribution” of the paper is located to the fact that the regional distribution system operator explores several alternatives on how to cope with the increased amount of EVs in combination to PV, district heating and flexible demand. As such, as it is clearly stated in the last paragraph in the introduction, the aim of this paper is to investigate several scenarios to achieve electrical power balance under high penetration of EVs in the Swedish distribution system. Specifically, this study is based on a real system in the northwest of Karlstad, Sweden, in Karlstads El- och Stadsnät’s power network. Several scenarios have been developed and simulated. The parameters that configure the scenarios are i) the number of grid connected EVs, ii) the level of charging rate, iii) the charging profile, iv) the flexible demand and v) the district heating versus heating using electrical units. Moreover, the case of residential PV systems is included when they are used for household demands. The purpose of the paper is to simulate the development of RES in the presence of different sources, such as PVs, district heating and ESD. Consequently, at first the impact of high EVs penetration to the power network is elaborated and second a better understanding of potential required measures and grid investments for the feasible operation of the RES are demonstrated.

- still the  method that used in the paper need more clarifications by adding flowchart and explain  what is the input and output of the model? How it works? How it will help us in the paper?

Response: Inputs are power profiles, consumers, storage and available energy sources. Output is the power balance according to the grid design. How it works: the equations involved in the method are analytically explained in the text and are given the appendix. The solution process (flowchart) is explained in the EnergyPLAN manual. Its presentation is outside of the scope of the paper and thus, the authors are kindly suggest to the reviewer to read the manual for details. 

- Clarify the size and resolution of the data? as the authors used real system what is the size the data that used and how they divide the data?

Response: The authors invite the reviewer to carefully read the section “2.2 The primary data of the simulation” especially the text just above Table 2 and the text next to Table 2.

- Results section, is still poor and repeated the same figures and tables without new information or new significant conclusion. The authors need to be critical about the results (figures and tables) and move the analysis close to each table or figure.

Response: The authors once again suggest to the reviewer to carefully read the results section because i) there are no figures in this section and ii) the tables are not the same since they present different results! We agree that more explanations on the findings should be presented and consequently, the section has been updated accordingly.

All sections have been revised as well as the following references were added in order to support the aim of the manuscript: 

- Volpe R., Frasca M., Fichera A., Fortuna L. “The role of autonomous energy production systems in urban energy network”, Journal of Complex Networks 2017, vol. 5 (3), pp. 461-472. DOI: https://doi.org/10.1093/comnet/cnw023

- Fichera A., Volpe R., Frasca M. “Assessment of the energy distribution in urban areas by using the framework of complex network theory”, International Journal of Heat and Technology 2016, vol. 34 (2), pp. S430-S434. DOI: https://doi.org/10.18280/ijht.34S234

- Gonzalez de Durana J., Barambones O., Kremers E., Varga L. “Agent based modeling of energy networks”. Energy Conversion and Management 82 (2014), pp. 308-319. DOI: 10.1016/j.econman.2014.03.018

-  Kremers E., Gonzalez de Durana J., Barambones O. “Multi-agent modelling for the simulation of a simple smart microgrid”. Energy Conversion and Management 75 (2013), pp. 643-650. DOI: 10.1016/j.econman.2013.07.050

Response to Reviewer#2:

Reviewer#2:

The authors have responded to all my queries and the overall response is satisfactory.

My suggestion is to improve the technical and scientific soundness of the manuscript as it seems to be very non-technically written and results are not well described.

Response: The authors appreciate reviewer’s opinion. The manuscript has been revised in all sections in order to improve the technical and scientific soundness. The introduction has been improved by adding more literature in order to support the background of the presented research. The results sections have been improved as well and the findings are clearly presented. The conclusions section has been rewritten accordingly.

Response to Reviewer#3:

The paper has been significantly improved. I feel also satisfied with the authort's responses to my last review. I would like to suggest the insertion of some papers dealing with the topic of renewable sources in networks, such as the case investigated by authors. It would be interesting to add some consideration about the penetration of PV and the autonomous management of energy from the final users, i.e. for the satisfaction of the energy demand or for the use of EV. 

Actually, as the authors themselves state in the abstract "The purpose is to examine the demands in order to cope with the needs that may arise and to create a better understanding of how renewable energy affects the power balance and future investments". Therefore, adding some literature that refer to distribution networks arising from renewable sources may fit to the scope.

Some suggestions are: 

- Volpe R., Frasca M., Fichera A., Fortuna L. “The role of autonomous energy production systems in urban energy network”, Journal of Complex Networks 2017, vol. 5 (3), pp. 461-472. DOI: https://doi.org/10.1093/comnet/cnw023

- Fichera A., Volpe R., Frasca M. “Assessment of the energy distribution in urban areas by using the framework of complex network theory”, International Journal of Heat and Technology 2016, vol. 34 (2), pp. S430-S434. DOI: https://doi.org/10.18280/ijht.34S234

- Gonzalez de Durana J., Barambones O., Kremers E., Varga L. “Agent based modeling of energy networks”. Energy Conversion and Management 82 (2014), pp. 308-319. DOI: 10.1016/j.econman.2014.03.018

-  Kremers E., Gonzalez de Durana J., Barambones O. “Multi-agent modelling for the simulation of a simple smart microgrid”. Energy Conversion and Management 75 (2013), pp. 643-650. DOI: 10.1016/j.econman.2013.07.050

Response: The authors appreciate reviewer’s opinion. The suggested literature has been included, as well as extra literature has been added to support the scope of the presented work. 

Moreover, I would also edit the figures. They are not suitable for the publication in a journal. It would be interesting to make them more appealing to the readers. 

Response: Although it not clear what kind of editing the reviewer suggests, the figures have been edited.

Reviewer 2 Report

The authors have responded to all my queries and the overall response is satisfactory.
My suggestion is to improve the technical and scientific soundness of the manuscript as it seems to be very non-technically written and results are not well described.

Author Response

(The authors gave the same response as above.)

Reviewer 3 Report

The paper has been significantly improved. I feel also satisfied with the authort's responses to my last review. I would like to suggest the insertion of some papers dealing with the topic of renewable sources in networks, such as the case investigated by authors. It would be interesting to add some consideration about the penetration of PV and the autonomous management of energy from the final users, i.e. for the satisfaction of the energy demand or for the use of EV. 

Actually, as the authors themselves state in the abstract "The purpose is to examine the demands in order to cope with the needs that may arise and to create a better understanding of how renewable energy affects the power balance and future investments". Therefore, adding some literature that refer to distribution networks arising from renewable sources may fit to the scope.

Some suggestions are: 

- Volpe R., Frasca M., Fichera A., Fortuna L. “The role of autonomous energy production systems in urban energy network”, Journal of Complex Networks 2017, vol. 5 (3), pp. 461-472. DOI: https://doi.org/10.1093/comnet/cnw023

- Fichera A., Volpe R., Frasca M. “Assessment of the energy distribution in urban areas by using the framework of complex network theory”, International Journal of Heat and Technology 2016, vol. 34 (2), pp. S430-S434. DOI: https://doi.org/10.18280/ijht.34S234

- Gonzalez de Durana J., Barambones O., Kremers E., Varga L. “Agent based modeling of energy networks”. Energy Conversion and Management 82 (2014), pp. 308-319. DOI: 10.1016/j.econman.2014.03.018

-  Kremers E., Gonzalez de Durana J., Barambones O. “Multi-agent modelling for the simulation of a simple smart microgrid”. Energy Conversion and Management 75 (2013), pp. 643-650. DOI: 10.1016/j.econman.2013.07.050

Moreover, I would also edit the figures. They are not suitable for the publication in a journal. It would be interesting to make them more appealing to the readers. 

Author Response

(The authors gave the same response as above.)
